Combination administration of alprazolam and N-Ethylmaleimide synergistically enhances sleep behaviors in mice with no potential CNS side effects

Zhu Siqing
Shi Jingjing
Zhang Yi
Chen Xuejun
Shi Tong shitongpla@163.com
Li Liqin llq969696@126.com
State Key Laboratory of NBC Protection for Civilian , Beijing , China
Banerjee Priyanka
Electronic publication date: 2024 May 7
Publication date: 2024
Volume: 12
Electronic Location ID: e17342
Received 2023 Dec 8; Accepted 2024 Apr 15
Copyright: ©2024 Zhu et al.
Copyright year: 2024
Copyright holder: Zhu et al.
License: This is an open access article distributed under the terms of the Creative Commons Attribution License, which permits unrestricted use, distribution, reproduction and adaptation in any medium and for any purpose provided that it is properly attributed. For attribution, the original author(s), title, publication source (PeerJ) and either DOI or URL of the article must be cited.
License URL: https://creativecommons.org/licenses/by/4.0/

Keywords: Combination, Alprazolam, KCC2, N-Ethylmaleimide, Enhanced, Sleep behaviors, Side effect

Funding: State Key Laboratory of NBC Protection for Civilian, Beijing SKLNBC2020-15 This work was supported by the State Key Laboratory of NBC Protection for Civilian, Beijing (Grant No. SKLNBC2020-15). The funders had no role in study design, data collection and analysis, decision to publish, or preparation of the manuscript.

==============================
Background

N-Ethylmaleimide (NEM), an agonist of the potassium chloride cotransporters 2 (KCC2) receptor, has been correlated with neurosuppressive outcomes, including decreased pain perception and the prevention of epileptic seizures. Nevertheless, its relationship with sleep-inducing effects remains unreported.

Objective

The present study aimed to investigate the potential enhancement of NEM on the sleep-inducing properties of alprazolam (Alp).

Methods

The test of the righting reflex was used to identify the appropriate concentrations of Alp and NEM for inducing sleep-promoting effects in mice. Total sleep duration and sleep quality were evaluated through EEG/EMG analysis. The neural mechanism underlying the sleep-promoting effect was examined through c-fos immunoreactivity in the brain using immunofluorescence. Furthermore, potential CNS-side effects of the combination Alp and NEM were assessed using LABORAS automated home-cage behavioral phenotyping.

Results

Combination administration of Alp (1.84 mg/kg) and NEM (1.0 mg/kg) significantly decreased sleep latency and increased sleep duration in comparison to administering 1.84 mg/kg Alp alone. This effect was characterized by a notable increase in REM duration. The findings from c-fos immunoreactivity indicated that NEM significantly suppressed neuron activation in brain regions associated with wakefulness. Additionally, combination administration of Alp and NEM showed no effects on mouse neural behaviors during automated home cage monitoring.

Conclusions

This study is the first to propose and demonstrate a combination therapy involving Alp and NEM that not only enhances the hypnotic effect but also mitigates potential CNS side effects, suggesting its potential application in treating insomnia.

Introduction

Sleep, an essential biological process, plays a critical role in maintaining good health, overall well-being, and public safety (Ramar et al., 2021). Inadequate sleep and untreated sleep disorders, such as insomnia, can have significant implications for both individual health and societal welfare (Lucena et al., 2020). Despite its prevalence and associated comorbidities like depression (Gebara et al., 2018), heart attacks (Arora et al., 2023), and Alzheimer’s disease (Lyon, 2019), managing insomnia effectively remains challenging (Khachatryan, 2021). Insomnia may arise directly from the disorder itself or indirectly from pain, depression, other sleep disturbances, or medication effects (Mayer et al., 2011). The COVID-19 pandemic has been associated with increased levels of mild insomnia symptoms, although not necessarily in moderate to severe cases (AlRasheed et al., 2022).

Among various treatments, benzodiazepine-receptor agonists (BzRAs) have the strongest empirical support. While these drugs were effective for short-term insomnia management, but their long-term efficacy was unclear, and they were associated with adverse effects such as morning sedation, dependence, tolerance, and rebound insomnia upon discontinuation (Morin & Benca, 2012).

Notably, a significant trend of benzodiazepine overdose, particularly in older patients susceptible to related adverse events, has been observed in China’s primary healthcare facilities (Fu et al., 2023). Alprazolam (Alp), an anxiolytic agent enhancing γ-aminobutyric acid (GABA) signaling by increasing the affinity of GABA for the GABAA receptor (Giordano et al., 2006), is a traditional benzodiazepine (Corser et al., 2023) that continues to be the primary choice for treating insomnia in clinical settings (Sellers et al., 1993), particularly in cases of anxiety-induced insomnia (Kurko et al., 2018). Information collected from the Food and Drug Administration revealed incidents of dementia linked to benzodiazepine, with Alp being responsible for 17% of these occurrences (Tampi & Tampi, 2014). Therefore, the debate over the widespread use of Alp and its adverse effects primarily revolves around GABA and GABAA receptor (Zhu et al., 2023).

Enhancing KCC2 function to increase chloride efflux and support GABAergic inhibitory mechanisms offers a potential approach to reducing neuronal excitability conditions. Varied levels of KCC2 expression and related chloride transport irregularities can directly impact the inhibitory actions of GABA, contributing to diverse neurological diseases such as epilepsy (McMoneagle et al., 2023), schizophrenia (Cherubini, Di Cristo & Avoli, 2021), and chronic pain (Aby et al., 2022). The most compelling evidence suggested that N-Ethylmaleimide (NEM), utilized as a KCC2 activator for numerous years (Conway et al., 2017; Chee, Kistler & Donaldson, 2006), primarily exerts its effect on KCC2 through the phosphorylation/dephosphorylation pathway (Jennings & Schulz, 1991).

Despite the proven benefits of KCC2 agonists in treating epilepsy and pain models (Shi et al., 2023), their potential to enhance sleep quality has not yet been investigated. This research gap necessitates evaluating the efficacy of combining BzRAs and KCC2 agonists in various sleep pharmacology models. Moreover, due to the lengthy and costly process of developing new medications, the innovative approach of combining already existing drugs has gained traction as a promising strategy for improving insomnia treatment (Welzel et al., 2021; Wesensten et al., 2005; Wright et al., 2016).

This study aimed to explore the effects of the combination of Alp and NEM on sleep in mouse, encompassing sleep behaviors tests, EEG/EMG analysis, and c-fos neuronal activation, as well as CNS safety pharmacology studied.

Materials & Methods

Experimental animals

C57BL/6J male mice (4-week, 19 ± 1 g) were obtained from the Beijing HFK Bioscience Co., Ltd. (Beijing, China). Prior to any experiments, the mice were kept in group housing (3–5 animals per cage) at 25 °C under a 12-hour light/dark cycle (lights turning on at 8 a.m.), with food and water provided ad libitum. The animal welfare and experimentation project were approved by the Animal Ethics Committee of State Key Laboratory of NBC Protection for Civilian (LAE-2022-03-001). At the end of experiments, the mice placed in the CO2 chamber were immediately euthanized by gradually increasing concentrations of CO2 at 10% displacement rate and continued to be ventilated for 2 min after cessation of breathing (Sengar et al., 2018). During the sleep behaviors tests, mice meeting the criteria of surviving surgery and exhibiting typical EEG/EMG signals were selected for the study, otherwise they were excluded. In other experiments, mice with regular spontaneous activity were included.

During the experiment, each mouse was assigned a temporary random number within the weight range and divided into each group (Li et al., 2019). The study was conducted in a double-blind fashion, where neither the experimental operator nor the data analyst could see each other’s results.

Compound preparation and administration

The Alp tablets was acquired from Yimin Pharmaceutical Co., Ltd, Beijing, China. NEM (97.0%) and sodium carboxymethylcellulose (CMC Na) were purchased from Sigma, USA.

Following our preliminary experiment and a random number table method, a total of 32 mice (eight mice per group) were randomly divided into the following treatment groups: (1) the control group treated with 1% CMC Na; (2) the Alp alone group treated with 1.84 mg/kg Alp in suspended with 1% CMC Na; (3) the NEM alone group treated with 1.0 mg/kg NEM in dissolved with 1% CMC Na; (4) the combination Alp and NEM (Alp@NEM) group treated with 1.84 mg/kg Alp and 1.0 mg/kg NEM in suspended with 1% CMC Na. All mice were intragastrically administered with a constant volume of 10 mL/kg body weight.

Sleep behaviors tests

The sleep behaviors tests were performed with reference to previously established protocols with slight improvements (Zhang et al., 2014; Li et al., 2021). Following different treatments, the mice were placed freely in a separate cage, where the ambient temperature was maintained at 37 °C. The mouse was gently rotated until it exhibited the signs of loss of righting reflex (LORR) and was unable to prostrate itself on limbs, a period defined as sleep latency. In addition, the mouse was unable to maintain LORR with at least three paws touching the bottom, a period defined as sleep duration. The sleep latency and sleep duration of each mouse were measured and recorded precisely with a chronometer (Fig. S1). Additionally, the sleep ratio was calculated as the percentage of mice that exhibited signs of LORR out of the total number of mice in each group.

EEG/EMG devices implantation surgery

Under 1.5% isoflurane inhalation anesthesia, mice were prepared for implantation of electrodes for electroencephalogram and electromyography (EEG/EMG) (Pinnacle Technologies, Lawrence, KS, USA) The electrodes were implanted in the skull of mice with reference to the previous surgical protocol (Sharma, Sahota & Thakkar, 2018; Jones et al., 2022). Shortly, three screw electrodes were implanted into the mouse’s skull and a head cap was placed to record EEG. The two flexible stainless Teflon-coated wires of the head cap were secured bilaterally into the trapezius muscle to record muscle activity (EMG). The entire setup was then fixed onto the skull using dental cement, with the wound closure achieved through suturing. Subsequently, the mouse was placed on a heated pad to aid recovery until it regained normal mobility. During the following week, postoperative anti-infective penicillin (40,000 units intraperitoneally per day) and analgesic meloxicam (4 mg/kg subcutaneously per day) were utilized.

Sleep recording and analysis

After one week habituation following surgery, the mouse’s head cap with a multichannel electrode pedestal was connected to an amplifier by a flexible cable and commutator. The acquired electrical signal was digitally converted using Sleep Sign (Kissei Comtec, Nagano, Japan). Wake-sleep status was scored automatically and manually in 4-second epochs. Wakefulness was defined by the presence of an asynchronous EEG and high-phasic EMG activity. Additionally, non-rapid eye movement (NREM) sleep was defined by a high-amplitude slow-δ wave EEG (0.5–4.0 Hz) together with a low EMG compared to wakefulness. Moreover, rapid eye movement (REM) sleep was defined by the presence of a high-power EEG θ wave (6.0–8.0 Hz) coupled with low EMG activity (suggestive of muscle atonia) compared to NREM. A chemical treatment was administered approximately 15 min before the EEG/EMG recording took place (Masuda et al., 2023).

Brain tissue preparation and immunofluorescence (IF)

After 30 min of administration, the mice in each group were quickly decapitated. Brain tissues were removed, fixed in 4% formaldehyde for 6 h, and then equilibrated in 20% sucrose in 1 × PBS for 48 h. Tissues were sectioned into 20 µm coronal sections using a Leica cryostat (CM1950; Leica Biosystems, Wetzlar, Germany) at −20 °C. The 20 µm frozen coronal sections were treated with 3.0% hydrogen peroxide for 25 min and rinsed by phosphate-buffered saline containing 0.1% Tween-20 (PBST). Sections were then blocked with bovine serum albumin containing 0.3% Triton x-100 for 2 h at 4 °C. After rinsing, sections were incubated with the primary antibody of rabbit anti-c-fos (1:1000 in PBS, ab222699, Abcam) overnight at 4 °C. Sections were rinsed and incubated with the anti-rabbit AlexaFluor-conjugated secondary antibody (1:500 in PBS, A32740; Thermo Fisher Scientific, Waltham, MA, USA) for 1 h at 37 °C and were treated with mounting medium containing DAPI (H-1500; Vectorlabs, Newark, CA, USA) before being sealed with a coverslip. The stained sections were then imaged using a confocal microscopy (Stellaris5 Leica, Wetzlar, Germany) with optical magnification of 20× and 40×. For each group, nine slices from three different mice were selected. In this experiments, c-fos expression in the ventrolateral preoptic nucleus (VLPO) and tuberomammillary nucleus (TMN) was quantified with ImageJ and the Cell Counter plugin. Cells with an immunofluorescence intensity exceedingly twice the background were classified as c-fos positive cells.

Continuous monitoring of laboratory mouse behaviors

The LABORAS system (Metris B.V., Hoofddorp, Netherlands) is a reliable and effective non-invasive tool that utilizes force measurement and pattern recognition technology. The automated home-cage behavioral experiment was conducted with a slight modification of a previously reported method (Quinn et al., 2003). Mice were administered with CMC Na, Alp (2.87 mg/kg for ED95 and 1.84 mg/kg for ED50), NEM (1.0 mg/kg), and a combination Alp and NEM (1.84 and 1.0 mg/kg, respectively). The spontaneous activity was monitored for a period of 24 h following awakening (when they were no longer to maintain LORR) in each group of mice (Fig. S2).

The effects of different treatments (vehicle, Alp with ED95 or ED50, NEM and the combination) on mice were manifested in terms of frequency and duration of behaviors including climbing, rearing, immobility, traveled distance and speed. Additionally, the distribution of activity paths within the cage was also visualized.

Statistics

Behavioral and imaging data statistics were analyzed using Student’s t-test or two-way ANOVA followed by Tukey’s multiple comparisons tests, as appropriate. All statistical analyses were conducted with GraphPad Prism version 9.0.2 for Windows (GraphPad Software, La Jolla, CA, USA). A significance threshold was set at p ≤ 0.01, with all values presented as Mean ± SEM.

Results

NEM can enhance Alp-induced sleep behaviors

The sleep latency, sleep duration, and sleep ratio of mice were evaluated after orally administering different doses of Alp (1.0, 1.2, 1.6, 2.0, 2.2 mg/kg) through the sleep behaviors tests. The results indicated that Alp induced sleep in C57 mice in a dose-dependent manner, with an ED50 of 1.84 mg/kg and an ED95 of 2.87 mg/kg (Table 1).

Subsequently, gradient doses of Alp (0.8, 1.0, 1.4, 1.6 mg/kg) less than 1.84 mg/kg were combined with gradient doses of NEM (0.5, 1.0, 1.6 mg/kg) (Yang et al., 2023) to assess the sleep effect, as illustrated in Fig. 1. Firstly, compared with Alp alone, the combination of gradient doses of NEM significantly decreased the sleep latency (Fig. 1A; p < 0.001), extended sleep duration of mice (Fig. 1B; p < 0.001), and increased the sleep ratio in different groups (Fig. 1C). Secondly, as the dosage of NEM increased, the sleep effect of NEM enhancing Alp appeared to become more pronounced. Finally, a dose–response study (Fig. 1D) illustrated that ED50 of sleep effect induced by different doses of NEM (0.5, 1.0, 1.6 mg/kg) were 1.41, 0.95, and 0.91 mg/kg, correspondingly. Surprisingly, there was no notable increase in efficacy when NEM doses exceeded 1.0 mg/kg. Therefore, 1.0 mg/kg NEM was selected for subsequent combination administration.

Notably, treatment with NEM alone did not exhibit any hypnotic effects (Fig. S3). Furthermore, the sleep behaviors tests indicated that a combination of Alp at 1.84 mg/kg (ED50) and NEM at 1.0 mg/kg aligns with the effect of administering 2.87 mg/kg Alp (ED95) alone, as confirmed in EEG/EMG recording analysis (Fig. S4).

Table 1 Effects of different doses of Alp on sleep (n = 8).

Dose (mg/kg)	Sleep ratio (%)	Sleep latency (min)	Sleep duration (min)	ED50/ED95 (mg/kg)	
2.2	87.5	31.43 ± 4.86	26.00 ± 4.55	1.84/2.87	
2.0	58.3	28.57 ± 6.88	24.86 ± 11.95	
1.6	42.9	36.67 ± 12.59	15.17 ± 5.74	
1.2	14.3	49.50 ± 0.71	9.00 ± 1.41	
1.0	0	IN	IN	
Notes.

IN indicated invalid sleep effect

Figure 1 NEM enhances the sleep-inducing effect of Alp.

(A) Sleep latency; (B) sleep duration; (C) sleep ratio; (D) dose–response curves for different dose combination of Alp and NEM. Data were expressed as mean ± SEM (n = 8) *p < 0.05 and **p < 0.01 vs. NEM at 0 mg/kg.

NEM significantly affects the architecture of Alp-induced sleep

The sleep quality is a crucial factor in assessing the effectiveness of any treatment aimed at improving sleep (Liu et al., 2020). Upon initial visual inspection, no discernible disparities in sleep quality were observed in mice that were administered with Alp alone compared to those combinedly administered Alp and NEM. Subsequently, post-treatment analysis of EEG/EMG recordings was conducted.

As shown in Fig. 2, compared to the control group, the duration of NREM (Fig. 2B; Alp vs. control, p = 0.0077; Alp@NEM vs. control, p < 0.0001) and REM (Fig. 2C; Alp vs. control, p < 0.0001; Alp@NEM vs. control, p < 0.0001) in the Alp and Alp@NEM groups was significantly prolonged, while the duration of awakening (Fig. 2A; Alp vs. control, p < 0.0001; Alp@NEM vs. control, p < 0.0001) was significantly reduced. Remarkably, the duration of REM in the Alp@NEM group was significantly longer than in the Alp group alone (Fig. 2C, p < 0.0001), suggesting that NEM enhanced the sleep-inducing effect of Alp by extending the REM portion of the sleep architecture.

Figure 2 The proportion of sleep phases.

(A) Wake, (B) NREM, and (C) REM in the total period, the distribution of activity paths (D), and the duration of immobility in mice following administration of vehicle, Alp, NEM, and a combination of Alp and NEM. The green region in (D) signifies a longer distance. Data were expressed as mean ± SEM (n = 6). *p < 0.01 and **p < 0.001 vs. control. ##p < 0.001 vs. Alp.

Furthermore, the spatial distribution of mice in the home cage over a 24-hour period following each treatment was illustrated in Fig. 2D. Mice that treated with either the vehicle or NEM displayed random and frequent exploration of the entire cage. In contrast, mice that treated with Alp or a combination of Alp and NEM demonstrated markedly decreased activity (Fig. 2E; Alp vs. control, p < 0.0001; Alp@NEM vs. control, p < 0.0001) in comparison to the control group, indicating a pronounced sedative-hypnotic effect. This trajectory plot was almost consistent with our visual observations (Fig. S3).

NEM inhibits the neurons activation in the TMN instead of activating VLPO nucleus neurons in the hypothalamus

In order to assess the activation of neurons in various hypothalamic regions following exposure to either Alp or NEM, c-fos expression, which is a proto-oncogene that serves as a marker for neuronal activity (Luo et al., 2018), was examined through immunostaining of the hypothalamus. The examination centered on the VLPO and TMN crucial hypothalamic nuclei implicated in regulating the sleep-wake cycle.

Expressions of c-fos were assessed in the TMN and VLPO in all four groups. Prior research has shown that c-fos levels were increased in the VLPO and decreased in the TMN during sleep (Scammell et al., 2001). Our research findings revealed that c-fos expression in the VLPO was elevated in both the Alp and Alp@NEM groups in comparison to the control group (Figs. 3A and 3C; Alp vs. control, p = 0.0031; Alp@NEM vs. control, p = 0.0097). There were no significant differences observed in the NEM group. On the other hand, there was a significant decrease in c-fos levels in the TMN (Figd. 3B and 3D; Alp vs. control, p = 0.0092; NEM vs. control, p < 0.0001; Alp@NEM vs. control, p < 0.0001) in the Alp, NEM, and Alp@NEM groups compared to the control group. More importantly, the expression of c-fos in the Alp@NEM group was notably lower compared to the Alp group (p = 0.0042).

Figure 3 Neuronal activation in the VLPO and TMN hypothalamic nuclei was observed following different treatments.

Quantification and IF images depicted c-fos positive cells in the VLPO (A, C) and TMN (B, D). Scale bar was one mm and 200 µm, respectively. Data were expressed as mean ± SEM (n = 3). *p < 0.01 and **p < 0.001 vs. control. # p < 0.01 vs. Alp.

These findings suggested that NEM may impact sleep patterns by diminishing neuronal activity in regions associated with wakefulness like the TMN, as opposed to regions promoting sleep like the VLPO.

No side effects of the combination administration of Alp (ED50) and NEM

The experiments demonstrated that combination administration 1.0 mg/kg NEM and 1.84 mg/kg (ED50) Alp was nearly as effective as administrating 2.87 mg/kg Alp (ED95) alone in inducing hypnotic effects (Fig. S3). However, the potential adverse effects were further investigated, including the impacts on general behaviors and well-being.

The results presented that the mice in the high-dose Alp (ED95) groups had significant differences in climbing duration (Fig. 4A; p = 0.0095), locomotion duration (Fig. 4C; p = 0.0001), immobility duration (Fig. 4D; p = 0.0026), and distance (Fig. 4E, p = 0.01) in comparison to the control group. Conversely, no significant differences were observed in the remaining groups. The results from this study showed that the administration of Alp (ED50), NEM, or a combination of both did not impact the spontaneous activity of locomotion. However, the highest dosage of Alp (ED95) did hinder locomotor activity, correlating with the common side effects of drowsiness reported in clinical studies (Su et al., 2022).

Figure 4 The effects of Alp, NEM, and the combination of Alp and NEM on spontaneous activity in mice.

The behavior was showed as (A) climbing duration, (B) rearing duration, (C) locomotion duration, (D) immobility duration, (E) total distance and (F) average speed. Data were expressed as mean ± SEM (n = 8). **p < 0.01 vs. control.

Additionally, the results revealed significant differences in food intake (Fig. 5B; p < 0.0001) and water consumption (Fig. 5C; p < 0.0001) between the control group and the high-dose Alp (ED95) groups. The results were consistent with the effects of dietary restriction (Haney et al., 1997) and increased water intake (Lobarinas & Falk, 2000) reported in Alp, and no significant differences were found in the other groups.

Figure 5 Changes of Alp, NEM, and the combination of Alp and NEM on (A) body weight loss, (B) food and (C) water intakes.

Data were expressed as mean ± SEM (n = 8). **p < 0.01 vs. control.

In summary, these findings suggested that the combined administration of Alp (ED50) and NEM did not adversely affect overall behaviors and welfare in mice.

Discussion

The findings of the current research demonstrated the sedative efficacy of Alp may be enhanced with the addition of NEM. The combination administration of Alp and NEM significantly decreased total wake time and increased the period of NREM and NEM in C57 mice. The expression of c-fos was significantly increased in the VLPO but decreased in the TMN. The data provided the first experimental evidence that combination administration of Alp and NEM may improve sleep by activating sleep-promoting region and deactivating awake-promoting region. In addition, the efficacy and safety of this administration strategy have been adequately verified.

This study demonstrated a synergistic effect of Alp and NEM on sleep behaviors, even though NEM alone did not cause any sleep-related changes in mice. Interestingly, it was observed that increased dosages of NEM had a negative impact on the overall improvement in sleep. To our knowledge, activation of the GABAA receptor by GABA or an agonist such as Alp initiates the influx of chloride into neurons, resulting in a decrease in membrane potential and the suppression of action potential generation (Zhang et al., 2016). Hence, it was theorized that the combination administration of Alp and NEM could potentially amplify the influx of chloride prompted by Alp, ultimately leading to a more potent inhibition of neuronal activity. Furthermore, the chloride gradient discrepancy induced by NEM may not be sufficient to provoke a central inhibitory impact. It is likely due to disruptions in the chloride balance within neurons located in the sleep-regulating brain regions, primarily influenced by alternative chloride transporters such as NKCC1 (Magloire et al., 2019; Ju et al., 2018). This also prompted us to consider whether inhibiting NKCC1, a chloride importer, could appear as a potential alternative for enhancing KCC2 and NKCC1 inhibitors (Moyon et al., 2021), which might also enhance Alp-induced sleep (Lorenzo et al., 2020).

Sleep architecture typically involves sleep/wake and REM/NREM patterns. Prior researches have indicated that hypnotic drugs typically alter total sleep duration and NREM rather than REM sleep (Shi et al., 2019; Kim et al., 2019; Yoon & Cho, 2018; Hajiaghaee et al., 2016). Nevertheless, longer REM duration has been linked with improved sleep quality (Feige et al., 2023; Mehta, Giri & Mallick, 2020). Our study found that the combined administration of Alp and NEM not only reduces wakefulness but also significantly promotes REM sleep, which potentially leading to important advancements in sleep pharmacology.

The expression of the c-fos protein were frequently utilized as markers for neuronal activation (Naik et al., 2015). The acute activation of melanin-concentrating hormone neurons in the lateral hypothalamus at the beginning of REM sleep has been found to prolong the duration of REM sleep (Jego et al., 2013). Endogenous adenosine in the TMN inhibits the histaminergic system through adenosine1 receptor and promotes NREM sleep (Oishi et al., 2008). These findings indicated that the activity of neurons in the TMN may could play a role in regulating the amount of time spent in REM sleep. Additionally, numerous studies indicated that neurons containing histamine, exclusively located in the TMN, play a key role in promoting wakefulness (Gerashchenko et al., 2004). Therefore, the c-fos results of this study suggested that reduction of TMN neuron activation by NEM may be closely related to the inhibition of histaminergic neurons, but further confirmation is still needed.

In behavioral experiments, the mice in the Alp@NEM group experienced enhancements in sleep without any significantly adverse effects, such as reduced activity often seen with elevated Alp doses. Two major disadvantages of long-term use of Alp were reduced GABAA receptor expression, which leading to potential tolerance (Yu et al., 2018), and impaired memory (Chowdhury et al., 2016). In a hypoxic mouse model (Yang et al., 2023), intraperitoneal injection of 10 mg/kg NEM has been shown to improve GABAergic function and memory performance. In this sense, NEM may serve as a complementary for Alp to address its limitations or challenges. However, insomnia treatments typically required a long-term medication (Verdoorn et al., 2019). Consequently, a combination therapy over an extended period is necessary to assess the potential side effects resulting from the buildup of Alp and NEM.

In summary, although we have presented for the first time a promising new medication strategy for insomnia that incorporates NEM while lowering the dose of Alp, and the efficacy and safety were well documented. However, the underlying molecular mechanisms remain elusive. Further investigations are required to assess alterations in levels of GABAA and KCC2 receptors, as well as neurotransmitters like histamine and GABA. In addition, our next step urgently needed to be carried out is to determine whether novel KCC2 agonists, such as Clp290 (Sullivan et al., 2021), exhibit a more potent impact than NEM on enhancing Alp-induced sleep. This investigation will contribute to the expanding field of research on the function of KCC2 agonists in regulating sleep and could have noteworthy implications for medical application in the future.

Conclusions

This research presented a novel approach for insomnia treatment by combining administration of 1.0 mg/kg NEM and Alp (ED50) for the first time. This strategy not only produced an efficient sleep-inducing effect comparable to administering Alp (ED95) dose but also decreased CNS side effects by lowering the dosage of Alp.

Limitations

While this research was the initial one to demonstrate that NEM could enhance the sleep benefits induced by Alp. Nonetheless, additional investigations at the molecular biology level were absent and should be carried out in future studies.

Supplemental Information

Figure S1 The moment when the mouse righting reflex disappears

Figure S2 Visual observation of mice in each group half an hour after administration (N=12)

Figure S3 The proportion of sleep phases

(A) wake, (B) NREM and (C) REM in the total time of the mice after treatment. Data were expressed as mean ± SEM (n = 8). *p < 0.05 and **p < 0.01 vs. control. # p < 0.05 vs. Alp-ED_50

Data S1 Raw data

Supplemental Information 5 Author Checklist

Abbreviations

Alp alprazolam

BzRAs benzodiazepine-receptor agonists

BSA bovine serum albumin

CMC Na carboxymethylcellulose

CNS central nervous system

EEG electroencephalogram

EMG electromyography

GABA Γ -aminobutyric acid

IF immunofluorescence

LORR loss of righting reflex

NEM N-Ethylmaleimide

NREM nonrapid eye movement

KCC2 potassium chloride cotransporters 2

PBST phosphate-buffered saline containing 0.1% Tween- 20

REM rapid eye movement

NKCC1 sodium potassium chloride cotransporter 1

TMN tuberomammillary nucleus

VLPO ventrolateral preoptic nucleus

Additional Information and Declarations

Competing Interests

Author Contributions

Animal Ethics

Data Availability

The authors declare there are no competing interests.

Siqing Zhu conceived and designed the experiments, prepared figures and/or tables, and approved the final draft.

Jingjing Shi conceived and designed the experiments, prepared figures and/or tables, and approved the final draft.

Yi Zhang analyzed the data, authored or reviewed drafts of the article, and approved the final draft.

Xuejun Chen performed the experiments, authored or reviewed drafts of the article, and approved the final draft.

Tong Shi analyzed the data, authored or reviewed drafts of the article, and approved the final draft.

Liqin Li performed the experiments, prepared figures and/or tables, and approved the final draft.

The following information was supplied relating to ethical approvals (i.e., approving body and any reference numbers):

The animal welfare and experimentation program were reviewed and approved by the Animal Ethics Committee of State Key Laboratory of NBC Protection for Civilian (LAE-2022-03-001).

The following information was supplied regarding data availability:

The raw measurements are available in the Supplementary File.

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
