# Peer review of "Combination administration of alprazolam and N-Ethylmaleimide synergistically enhances sleep behaviors in mice with no potential CNS side effects"

_PeerJ, doi:10.7717/peerj.17342_

## Round 0.1 · original submission · Major Revisions

Please respond to the reviewer's comments and send the revised manuscript.

**Language Note:** The review process has identified that the English language must be improved. PeerJ can provide language editing services - please contact us at [email protected] for pricing (be sure to provide your manuscript number and title). Alternatively, you should make your own arrangements to improve the language quality and provide details in your response letter. – PeerJ Staff

·

Basic reporting

I have reviewed the manuscript titled "Combination administration of Alprazolam and N-Ethylmaleimide synergistically enhances sleep behaviors in mouse with no potential CNS side effects". The paper aims to provide a comprehensive understanding of the impact of NEM and Alp in treating insomnia. This is a significant topic, and a comprehensive review is essential for researchers interested in this area.

However, the manuscript could benefit from the addition of a few molecular biology techniques to support the research. It also needs to be structured more appropriately, and a few comments addressed. Specifically, the manuscript should discuss the strong interaction of NEM and Alp administration in treating insomnia and achieving an effective sleep-inducing effect. It should include a clear methodology, results, discussion, and graphical representation. This will require major and thorough revisions.

Experimental design

To improve the manuscript, the authors should follow the following suggestions:

1. Authors should add appropriate keywords that support the research in the abstract section and arrange them alphabetically.
2. Authors should mention a list of full abbreviations.
3. Authors should rewrite the introduction in 3-4 paragraphs and include the mechanism of action of NEM and Alp.
4. Authors should correct the ad libitum in an italic font.
5. Authors should reframe the paragraph on lines 88-92.
6. Authors should check the toxicity level of NEM and Alp in the mice and add the justification for using therapeutic doses.
7. Authors should rewrite the statement on lines 135-136.
8. Authors should change the magnification of 20X and 40X instead of 200X and 400X.
9. Authors should explain why the authors used two types of T-tests on line 166 and what the difference is between Student’s t-test and Welch’s t-test.
10. Authors should rewrite the “NEM can enhance Alp-induced sleep behavior” result section on line 172 and provide justification for the last paragraph.
11. Authors should explain the full form of the VLPO and TMN on line 214.

Validity of the findings

The authors should improve the discussion section and provide more justification for the study. The authors should expand upon the knowledge gap of this study. Authors should rewrite the conclusion section with strong evidence that reflects the research and include a brief future perspective section separately.

Reviewer 2 ·

Basic reporting

The study investigates the synergistic effects of N-Ethylmaleimide (NEM) on the hypnotic action of Alprazolam (Alp) in mice, revealing that a combined administration significantly enhances sleep duration and quality, specifically by increasing REM sleep, without eliciting central nervous system side effects. The findings indicate that NEM augments Alp's efficacy by inhibiting wakefulness-associated neuronal activity in the brain, suggesting a potential new treatment for insomnia that reduces necessary Alp dosages and associated risks.

The manuscript is overall well written, but there are still some typos that can be corrected. For example, “ After different treatment, the mice were freely placed” should be “after different treatments”. I suggest the authors to go through the manuscript again carefully.

Experimental design

The overall experiment design is clear. The authors reported sleep ratio in figure 1. I would like to see a clear definition of sleep ratio in figure 1.

Also, out of curiosity, have the authors measured the core body temperature of the mice after administering these drugs. The relative immobility and other behaviors reminded me of a torpor like state in mice (See https://doi.org/10.1038/s41586-020-2387-5). A key difference between sleep and torpor is the amount of body temperature drop.

Validity of the findings

For Fig1b where is the data for the 0mg/kg NEM for the 0.8 and 1.0 Alp conditions?
The authors should include NEM label in the legend for better clarity.

The data in fig2b showed that the combination of AIP and NEM also increases NREM sleep time. Therefore, the title of the results section, “Combined administration of Alp and NEM significantly increase NREM duration in sleep architecture”, does not seem to accurately reflect the findings. I understand the authors are trying to make the point that the combination may have a more significant effect on REM sleep. However, the difference is not clear to me in this context. The authors should consider reporting the specific p values instead of only a range. It is generally a good idea to include specific p values throughout the article as well.

Also, Fig2D is not convincing to me either, especially between Alp or NEM conditions. Both seem to be active. If the authors choose to show this, they need to quantify the measurements.

For figure 3, please show the brain coordinates (+/- bregma) of the slice used. Also, please elaborate on how to make sure we are looking at the same brain areas across different conditions. The example slices shown here seem to vary in the anterior and posterior axis.

---

## Round 0.2 · accepted · Accept

Thank you for addressing all the reviewer's comments and submitting the revised manuscript.

·

Basic reporting

No comments

Experimental design

No comments

Validity of the findings

No comments

Reviewer 2 ·

Basic reporting

In the rebuttal letter, the authors mentioned that the title of the article has changed to "NEM significantly affects the architecture of Alp-induced sleep". However, I did not see this get reflected in the actual manuscript.

Experimental design

I have no more comments.

Validity of the findings

I have no more comments.

Additional comments

I have no more comments.